# COMD: Training-free Camera Motion Transfer With Camera-Object Motion Disentanglement

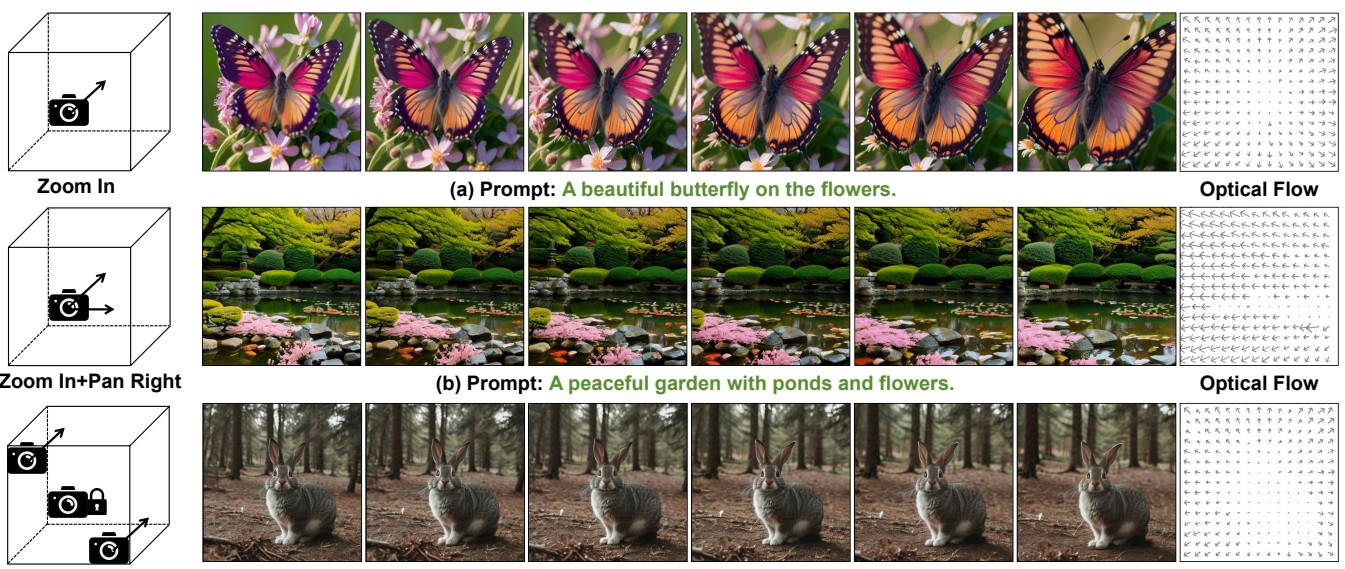

**Figure 1: Flexible and diverse camera motion control of our training-free COMD. COMD can control (a) one camera motion or (b) combine several camera motions in one video. Moreover, COMD enables control of different camera motions in different regions, which can achieve professional Dolly Zoom with zooming motions in the background and fixed motion in the foreground (c).**

## ABSTRACT

The emergence of diffusion models has greatly propelled the progress in image and video generation. Recently, some efforts have been made in controllable video generation, including text-to-video, image-to-video generation, video editing, and video motion control, among which camera motion control is an important topic. However, existing camera motion control methods rely on training a temporal camera module, and necessitate substantial computation resources due to the large amount of parameters in video generation models. Moreover, existing methods pre-define camera motion types during training, which limits their flexibility in camera control, preventing the realization of some specific camera controls, such as various camera movements in films. Therefore, to reduce training costs and achieve flexible camera control, we propose COMD, a novel training-free video motion transfer model, which disentangles camera motions and object motions in source videos, and transfers the extracted camera motions to new videos. We first propose a one-shot camera motion disentanglement method to extract camera motion from a single source video, which separates the moving objects from the background and estimates the camera motion in the moving objects region based on the motion in the background by solving a Poisson equation. Furthermore, we propose a few-shot camera motion disentanglement method to extract the common camera motion from multiple videos with similar camera motions, which employs a window-based clustering technique to extract the common features in temporal attention maps of multiple videos. Finally, we propose a motion combination method to combine different types of camera motions together, enabling our model a more controllable and flexible camera control. Extensive experiments demonstrate that our training-free approach can effectively decouple camera-object motion and apply the decoupled camera motion to a wide range of controllable video generation tasks, achieving flexible and diverse camera motion control.

*ACM MM, 2024, Melbourne, Australia*
© 2024 Copyright held by the owner/author(s). Publication rights licensed to ACM.
ACM ISBN 978-x-xxxx-xxxx-x/YY/MM
https://doi.org/10.1145/nnnnnnn.nnnnnnn

## CCS CONCEPTS

• **Computing methodologies** → **Computer vision**.

## KEYWORDS

Video Generation, Video Motion, Camera Motion, Disentanglement

# 1 INTRODUCTION

In recent years, the rapid development of generative models [20, 27] has led to significant advancements in the field of image and video generation. Among video generation, diffusion models [3, 7, 18, 30] have emerged as powerful tools for generating high-quality videos with high diversity. Meanwhile, the demand for controllable video generation has grown significantly, especially in applications such as film production, virtual reality, and video games, where researchers have devoted much effort to controllable generation tasks including text-to-video generation [6, 7, 18, 30], image-to-video generation [3, 18], video motion control [5, 8, 33, 44], and video editing [1, 26]. Since video is composed of a sequence of images with consistent and fluent motions, the control of video motion has become an important topic in controllable video generation.

For video motion control, **1)** most of the existing methods [1, 5, 8, 42] focus on modeling the *object motion* and use trajectory or a source video to guide the movement of the objects, but usually lack the ability to model the camera motion. **2)** To enable the control of the *camera motion*, AnimateDiff [18] trains temporal LoRA modules [22] on a collected set of videos with the same camera motion. To control different camera motions using one model, MotionCtrl [40] labels a large number of videos with corresponding camera pose parameters to train a camera motion control module. In contrast, Direct-a-video [44] utilizes a self-supervised training process by manually constructing camera motions along x, y, and z axis, reducing the training resources to some extent. However, all the existing camera motion control methods rely on training a temporal camera module to control the camera motion, which poses a significant requirement to the computational resources due to the large number of parameters in video generation models. Moreover, these methods can only achieve simple camera motion control and cannot handle some complex and professional camera motions in films, such as Dolly Zoom (zoom in or out the camera while keeping the object still) and Variable-Speed Zoom (zoom with variable speed).

To achieve complex camera motion control and reduce the training costs, we propose COMD, a novel **training-free camera motion transfer model**, which disentangles camera motions and object motions in source videos and then transfers the extracted camera motions to new videos. Firstly, we observe that the temporal attention maps in diffusion-based video generation models contain the information of video motions, and find that the motions are composed of two motion types, camera motions and object motions. We then propose two methods to disentangle the camera motions and object motions in temporal attention maps. 1) In **one-shot camera motion disentanglement**, we decompose camera and object motions from a single source video. We regard the motion in the background as only containing camera motion, while the motion in the foreground as containing both camera and object motions. We employ a segmentation model to separate the moving objects and background regions, and then predict the camera motion in foreground region from background motion by solving a Poisson equation. 2) To further enhance the disentanglement ability, we propose a **few-shot camera motion disentanglement** method to extract the common camera motion from several videos with similar camera motions, which employs a novel window-based

clustering method to extract the common features from temporal attention maps of multiple videos. Finally, we investigate the additivity and positional composition ability of camera motions, and propose a camera motion combination method to achieve flexible camera control, which can enable combining different kinds of camera motions into a new motion, and apply different camera motions in different regions.

Extensive experiments demonstrate the superior performance of our model in both one-shot and few-shot camera motion transfer. With the camera motion combination and the disentanglement between the camera motion and position, our model substantially improve the controllability and flexibility of camera motions.

The main contributions can be summarized as follows:

- We propose COMD, a training-free camera motion transfer method based on **C**amera-**O**bject **M**otion **D**isentanglement, which can transfer the camera motion from source videos to newly generated videos.
- We propose a novel one-shot camera-object motion disentanglement method. By separating the moving objects and the background regions and estimating the camera motion in the moving objects region by solving a Poisson equation, our model can effectively disentangle the camera motion from object motion in a single video.
- We further propose a few-shot camera-object motion disentanglement method, which employs a novel window-based clustering method to extract the common camera motion from several given videos with similar camera motions, effectively dealing with scenarios with overly complex and diverse object motions.
- We propose a camera motion combination method to achieve flexible camera motion control, which enables the model to combine different camera motions into a new motion and apply different camera motions in different regions.

# 2 RELATED WORK

## 2.1 Text-to-Video Generation

Generative models have rapidly advanced and achieved tremendous success in text-driven video generation tasks, which mostly rely on generative adversarial networks (GANs) [29, 36, 38, 47] and diffusion models [3, 4, 6, 7, 16, 18, 21, 30] Among these methods, diffusion models have emerged as a powerful tool due to their ability to generate diverse and high-quality contents. Early text-driven video generation models [19, 21, 30] perform diffusion in pixel space, requiring cascaded generation and significant computational resources to generate high-resolution videos. Recent research papers have implemented diffusion in the latent space [3, 4, 18, 28, 37, 49], achieving high-quality and long-duration video generation. Additionally, researchers are exploring more controllable video generation approaches. For instance, [9, 11, 17] introduce spatial and geometric constraints to generative models, [41] generates videos of desired subject, and [8, 40] govern motion in generated videos. These methods enable users to finely control various attributes of videos, resulting in generated outcomes that better align with user preferences and requirements.

## 2.2 Motion Controllable Video Generation

**Object Motion Control.** Many researches [23, 24, 32, 33, 41, 43, 48] have been conducted to control object motions to better align with user preferences. Some methods [5, 44] enable users to control the motion of objects by dragging bounding boxes, while some other works [23, 40] allow control over the trajectory of the object. Video-Composer [39] provides global motion guidance by conditioning on pixel-wise motion vectors. Besides, some video editing methods [1, 10, 26, 33] also enable motion editing through text-driven or manually specified motions, which requires motion consistency between adjacent frames. In summary, all these works focus more on controlling the object motions rather than camera motions, which operates at a local, high semantic level.

**Camera Motion Control.** There have been relatively few researches in camera motion control. AnimateDiff [18] employs temporal LoRA modules [22] trained on a collected set of videos with similar camera motion. Thus a single LoRA module is capable of controlling only a specific type of camera motion. MotionCtrl [40] constructs a video dataset annotated with camera poses to learn camera motions, but requires substantial manual effort. Direct-a-video [44] adds camera motion along coordinate axes to existing videos, which can reduce annotation costs. However, all of these works require fine-tuning pretrained video generation models, consuming a large amount of computation resources and limiting the style of camera motion to the training data. In contrast, our model enables flexible camera motion control with any target camera motions without re-training the model, which brings a much wider application for camera control in video generation.

## 3 METHOD

Our COMD model aims to **disentangle the camera motion and object motion** in a single or several videos, and then **transfer the disentangled camera motion** to the newly generated videos. We first observe that the temporal attention maps in diffusion-based video generation models contain the information of videos motions, and find that the motion are composed of two motion types, camera motions and object motions. We then propose two methods to decompose the temporal attention map $Attn$ into object motion $Attn^o$ and camera motion $Attn^c$, as shown in Fig. 2. By substituting the temporal attention map with the temporal attention of the target camera motion, we can enable the video generation models to generate videos with the desired camera motion.

Specifically, to disentangle the camera motion from the object motion, we propose to extract the camera motions from either a single video or a few (5-10) videos. **1)** In **one-shot camera motion disentanglement**, we aim to extract camera motion from a single video (Fig. 2 top). Considering the motion in background region only contains camera motion, while motion in the foreground region contains both camera motion and object motion, we first separate background and foreground regions. We employ SAM [25] to segment the moving objects, and decompose the given video into moving object region $M$ and background region $\tilde{M} = 1 - M$. Then we regard the motion in the background region $\tilde{M}$ as only containing camera motion. With the observation that the camera motion is smooth and continuous, and the neighboring pixels share similar motions [14, 15, 46, 50], we construct a Poisson equation

to estimate the camera motions in the moving objects region $M$ based on the given camera motions in the background region $\tilde{M}$, achieving camera-object motion disentanglement for a single video.

**2)** When the object motions are too complex to disentangle from a single video, we propose a **few-shot camera motion disentanglement** method to extract common camera motion from $m$ (5-10) videos with similar camera motions (Fig. 2 bottom). To extract common camera motion of $m$ videos, we regard the common feature of the temporal attention maps of these videos as the feature of the common camera motion. We then propose a window-based clustering method for each pixel of the temporal attention map to extract the common camera motion and filter out outliers. Specifically, we regard the neighboring pixels in a $k \times k$ window share similar camera motions and cluster the $k^2$-neighboring pixels of each pixel in the $m$ temporal attention maps with DBSCAN clustering method, where the centroid of the largest cluster can be used to represent the common camera motion.

Finally, we investigate the additivity and positional composition ability of camera motions. We propose a camera motion combination method to achieve flexible camera motion control, which can combine different camera motions into a new motion and apply different camera motions in different regions, substantially improving the controllability and flexibility of camera motions.

## 3.1 Camera Motion Extraction Based on Temporal Attention

**Preliminaries of temporal attention module.** Most of the current video generation models [3, 4, 18] are built on a pretrained text-to-image diffusion model [27], which employs spatial attention module to model the image generation process. To extend the image generation models to generate videos, temporal attention module [4, 18] is proposed to enable the pretrained image generation models with the ability to model the temporal relationship between each frame of the video. Specifically, the temporal attention mechanism is a self-attention module, which takes the feature map $f_{in}$ of $t$ frames ($b \times t \times c \times h \times w$) as input, and reshapes it to a $(b \times h \times w) \times t \times c$ feature map $f$. Then, a self-attention module is employed to capture the temporal relationships between $t$ frames, and output a feature map with temporal relationships between each frame, which is formulated as follows:

$$Attn = Softmax(\frac{QK^T}{\sqrt{c}}),\ f_{out} = AttnV, \qquad (1)$$

where $Q = W_Q f$, $K = W_K f$ and $V = W_V f$, and $W_Q$, $W_K$ and $W_V$ are learnable query, key and value matrices.

**Extracting motion information from temporal attention map.** UniEdit [1] found that the temporal attention modules model the inter-frame dependency and motion information[1], and use the temporal attention for video motion editing tasks, where the global motion of video is edited guided by text. However, it lacks a deep analysis of how the temporal attention module models the inter-frame dependency. In this paper, we find that the attention maps $Attn$ of the temporal attention layer are composed of **two motion types**, which are **camera motions** and **object motions**. We propose two methods to decouple motion in temporal attention

---

[1]Our experiments also validate this finding, shown in #Suppl.

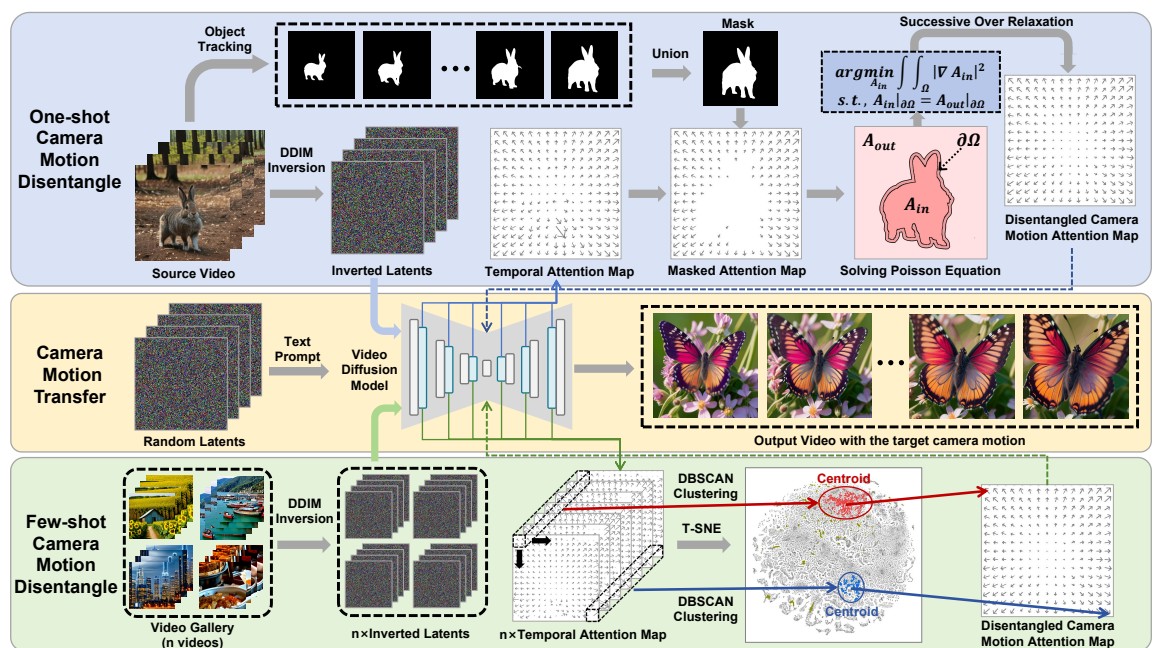

**Figure 2: *Main framework of our method:*** Our model can extract the camera motion from a single video or several videos that share similar camera motions. **1) *One-shot camera motion disentanglement:*** We first employ SAM [25] to segment the moving objects in the source video and extract the temporal attention maps from the inverted latents. To disentangle the camera and object motions, we mask out the object areas of the attention map and estimate the camera motion inside the mask by solving a Poisson equation. **2) *Few-shot camera motion disentanglement:*** we extract the common camera motion from the temporal attention maps of several given videos. For each position $(x, y)$, we employ all of its k-neighboring attention map values across each video for clustering. Then, we use the centroid of the largest cluster to represent the camera motions in position $(x, y)$.

map into camera and object motions (Sec. 3.2 and 3.3), where we disentangle the temporal attention map $Attn$ extracted from a video into camera motion attention $Attn^c$ and object motion attention $Attn^o$. After decoupling camera motion from object motion, we can easily transfer camera motion from a source video $v_s$ to a target video $v_t$, by replacing the temporal attention map of $v_t$ with the temporal attention map $Attn^c_s$ that corresponds to the camera motion of $v_s$:

$$f_{out} = Attn^c_s V. \qquad (2)$$

### 3.2 One-shot Camera Motion Disentanglement

In this section, we propose a method to disentangle camera motions from object motions in a single source video. A video can usually be divided into two parts: the foreground region and the background region. Considering the motion in background region mainly contains camera motion, while the motion in foreground region contains both camera motion and object motion, we first extract camera motion in background region, and then predict camera motion in foreground based on the background camera motion.

Specifically, we first employ segment-anything model to segment the moving objects and the background, and then take the temporal attention map from the background region as the camera motions. Based on the observation that the camera motions are continuous and the neighboring pixels have similar camera motions, we construct a Poisson equation to estimate the camera motions inside the

moving object region based on the camera motions outside, thereby achieving the camera-object motion disentanglement.

**Obtaining temporal attention map of a video by DDIM inversion.** First of all, to obtain the temporal attention map of the source video $v_s$, we apply DDIM inversion on the source video to invert it into a $T$-step latent $x_T$. Then, by denoising $x_T$ with the video diffusion model, we obtain a series of attention maps $\{Attn_T, Attn_{t-1} \cdots Attn_1\}$ in different timesteps. Different from the spatial attention maps (in spatial attention modules), which model different spatial relationships in different timesteps, the temporal attention maps model *the temporal motion* of the video, and we find they are similar in different timesteps. Therefore, we can use one representative temporal attention map $Attn = Attn_t$ at timestep $t$ to model the temporal motion, which can effectively reduce the computation resources to $\frac{1}{T}$ of using all timesteps. We adopt a medium timestep $t$, since when $t$ is large, there are too many noises in the video feature; while when $t$ is small, the denoising has almost been completed and the overall motion has already been determined, thus the motion information in the temporal attention map at small $t$ is not sufficient.

**Extracting camera motion in background region.** With the obtained temporal attention map $Attn$ from the source video, we employ segment anything model ($SAM$) to obtain the mask of the moving objects in each frame $M_i = SAM(v_i), i = 1, \cdots, t$, where $v_i$ denotes the $i$-th frame of the source video $v_s$. Then, we merge the masks of $t$ frames into one mask $M = U(M_1, M_2 \cdots M_t)$. Since

the motion in the background region mainly comes from the camera motion, we regard the masked temporal attention map in the background region $Attn_m = Attn \odot (1 - M)$ as the camera motion attention map that only controls the camera motion. Although currently the masked attention map $Attn_m$ has no value inside the moving objects mask $M$, we can estimate the camera motion inside the mask based on the camera motion outside. To estimate the camera motion inside the mask $M$, we transform the motion estimation problem into solving a Poisson equation, which is introduced below.

**Predicting camera motion in foreground region.** Video processing tasks such as video compression, optical flow estimation, and video interpolation, share a common assumption that the changes between video frames are smooth and continuous [14, 15, 46, 50], and the motions of the pixels in a local neighborhood are similar. Based on this assumption, we posit that the **camera motion** is also **continuous and has local coherence**, *i.e.,* the camera motions in a local region are almost the same. Therefore, we assume the gradient of the camera motion attention map inside the mask region is quite small, and the values of the attention map on both sides of the mask boundary are almost the same. Denote the camera motion attention map inside the mask $M$ as $A_{in}$ (to be estimated), and the camera motion attention map outside the mask as $A_{out}$ (which we already have $A_{out} = Attn_m$). And we denote the positions of each pixel inside the mask as $\Omega \in \mathcal{R}^2$, and the mask boundary as $\partial\Omega$. Then, we have $\nabla A_{in} \approx 0$, and $A_{in}|_{\partial\Omega} = A_{out}|_{\partial\Omega}$. Since we already know $A_{out}$, we can estimate $A_{in}$ by solving the following optimization problem:

$$A_{in}^* = \underset{A_{in}}{argmin} \int \int_{\Omega} \|\nabla A_{in}\|^2. \qquad (3)$$
$$s.t. \, A_{in}|_{\partial\Omega} = A_{out}|_{\partial\Omega}.$$

Therefore, the camera-motion estimation problem is converted into a Poisson blending problem. By setting the gradient inside the mask to be 0, we can employ Successive Over Relaxation algorithm [45] for Poission Blending to find the optimal solution $A_{in}^*$. Finally, we obtain the complete camera motion attention map $Attn^c = \{A_{in}^*, A_{out}\}$, which is disentangled with the object motion. With the disentangled $Attn^c$, we can employ the camera motion transfer method in Sec. 3.1 to transfer the camera motion from a single source video to target videos.

## 3.3 Few-shot Camera Motion Disentanglement

When the object motions are overly complex to disentangle, *e.g.,* moving objects may occupy nearly all the pixels, it may be difficult to disentangle camera motion and object motion from a single video. To improve the disentanglement performance for videos with complex object motions, we relax the input conditions from one shot to few shot. I.e., we aim to extract the common camera motion from several videos $\{v_1, v_2 \cdots v_m\}$ with similar camera motions.

**Extracting common feature in temporal attention as common camera motion.** In Sec. 3.2, we decompose the temporal attention maps of a single video into camera motion and object motion. Since the given $m$ videos $\{v_1, v_2 \cdots v_m\}$ share similar camera motions, we regard the common feature of the temporal attention maps as the feature of camera motion. Therefore, we calculate common camera motion by extracting a common feature from the

temporal attention maps of $m$ videos. Since the motion at different locations may be different (*e.g.,* zoom in/out), we model the motion at pixel level. Denote the temporal attention map of each video as $\{A_1, A_2 \cdots A_m\}$, where $A_i \in \mathcal{R}^{W \times H \times t \times t}$ and $t$ is the number of frames. For each pixel $(x, y)$ in video $v_i$, we denote its motion as $A_i(x, y) \in \mathcal{R}^{t \times t}$. Next, we aim to extract the common feature for each pixel $(x, y)$ from $m$ temporal attention maps.

**Local coherence assumption for camera motion.** To extract the common feature for each pixel $(x, y)$, only using the attention values at the location $(x, y)$ in $m$ temporal attention maps may not be adequate, especially when the object motions in the given $m$ video are complex and diverse. Therefore, based on the assumption of local coherence, we regard that the neighboring pixels in a window centered at pixel $(x, y)$ share similar camera motion as the center pixel. In other words, we extract the common camera motion for the pixel $(x, y)$ by considering the attention values of neighboring pixels in a $k \times k$ window $\mathcal{N}_k(x, y)$ in each of the $m$ temporal attention maps ($m \times k^2$ pixels in total), whose attention values form a tensor $\mathcal{A}(x, y) = \{A_i(\mathcal{N}_k(x, y)), i = 1 \cdots m\} \in R^{m \times k^2 \times t \times t}$.

**Extracting common camera motion by window-based clustering.** For each pixel $(x, y)$, to extract the common camera motion from the attention values $\mathcal{A}(x, y)$ in its $k \times k$ neighboring window, we first reshape the attention values $\mathcal{A}(x, y)$ to $R^{(m \times k^2) \times (t \times t)}$. We then employ t-SNE [35] to reduce the dimension from $(t \times t)$ to 2, for better clustering in the subsequent steps. After dimension reduction, we compute the centroid of the $m \times k^2$ pixels as the representation of the common camera motion. Directly computing the mean value of all the $m \times k^2$ pixels is a possible solution to compute the centroid, but has inferior accuracy of the extracted motion when the camera motions in some of the samples are severely entangled with object motion. Therefore, we employ DBSCAN [12] to cluster all the pixels, which can effectively distinguish the outliers. After clustering, we have $n_c$ clusters, with each cluster containing part of the attention values. We regard the centroid of the largest cluster as the common camera motion, since it is the most common motion among the $m \times k^2$ pixels. With the extracted camera motion map $Attn^c$, we can transfer the camera motions to new videos.

## 3.4 Camera Motion Combination

**Camera motion combination.** In Sec. 3.2 and 3.3, we extract the camera motion $Attn^c$ from a single or several videos. These camera motions can work separately by transferring one extracted camera motion to a target video. One natural question is whether we can combine different camera motions to enable a more complex and flexible camera motion control. To achieve this, in this section, we explore different ways to combine camera motions, which enables **1)** combining different camera motions into a new motion; **2)** applying different camera motions in different areas; and **3)** preserving part of the contents while transferring the camera motion.

**Additivity of the camera motions.** We first explore how to combine different camera motions together. We are delighted to discover that the camera motions extracted from Sec. 3.2 and 3.3 are additive. By adding the attention maps $\{Attn_i^c\}_{i=1}^n$ corresponding to different camera motions, we can obtain a new camera motion that includes all the combined camera motions at the same time. And by assigning different weights $\{w_i\}_{i=1}^n$ to different camera motions,

**Figure 3: The comparison on one-shot and few-shot camera motion transfer with AnimateDiff+Lora [18] and MotionCtrl [40]. AnimateDiff+Lora tends to overfit to the training data while MotionCtrl suffers from shape distortions and logical inconsistencies when controlling camera motion, even though it is trained on large-scale data. In contrast, our COMD generates high-quality videos with accurate camera motions.**

we can control the intensity of each camera motion by:

$$Attn_{new}^c = \sum_{i \in Sub(\{1 \cdots n\})} w_i \times Attn_i^c, \qquad (4)$$

where $Sub(\{1 \cdots n\})$ is an arbitrary subset of $\{1 \cdots n\}$.

**Position-specified motion transfer.** The camera motion transfer methods in previous sections can only transfer the camera motions in a global manner, while lacking the ability to transfer the camera motions in a local region. Therefore, to enable our model with the ability to control the camera motions in a local manner, we propose a segmentation-based local camera motion control method. We segment local regions by SAM, and assign different camera motions to different local regions of the generated video, by applying the mask $M_i$ on the camera motion attention map $Attn_i^c$ as follows:

$$Attn_{new}^c = \sum_i M_i \odot Attn_i^c. \qquad (5)$$

**Local content-preserving camera motion transfer.** To better preserve specific content within the target video, we first utilize SAM to segment the object region $M$ we aim to keep unchanged and then modify the temporal attention calculation. We find that in diffusion-based video generation models, the appearance and motions are well disentangled in the temporal attention modules, where the temporal attention maps represent the temporal motions, while the Value $V$ represents the appearance. Therefore, when we need to transfer the camera motions from a source video $v_s$ to a target video $v_t$ while keeping the appearance in region $M$ of $v_t$ unchanged, we modify the temporal attention calculation by keeping the Value inside $M$ the same as the Value $V_t$ of the target video, and substituting the temporal attention map by the camera motion attention map $Attn_s^c$ of the source video, which can be formulated as follows:

$$V' = V_t \odot M + V \odot (1 - M), \ f_{out} = Attn_s^c V'. \qquad (6)$$

## 4 EXPERIMENTS

### 4.1 Implementation Details

**Experiment details and hyperparameters.** In our experiments, we adopt AnimateDiff [18] as the baseline method for motion disentanglement and control, which is one of the state-of-the-art text-to-video models. The generated video size is $512 \times 512$, with each video composed of 16 frames with 8 FPS. When generating videos, we employ 25-step DDIM [31] for inference and choose the temporal attention maps in the 15-th step to extract the camera motions. Moreover, for few-shot camera motion extraction, we compute the neighborhood size $k$ by $k = \lceil \frac{size}{16} \rceil \times 2 + 1$, where $size$ is the width and height of the temporal attention maps.

**Evaluation metrics.** To evaluate the generation quality, diversity and camera motion accuracy, we employ three evaluation metrics: 1) FVD [34]: Fréchet Video Distance measures the quality and authenticity by calculating the Fréchet distance between real and generated videos; 2) FID-V [2]: Video-level FID uses a 3D Resnet-50 model to extract video features for video-level FID scoring, measuring the quality and diversity of the generated videos; and 3) Optical Flow Distance [13] assesses the camera movement accuracy by computing distance between the flow maps from the generated and ground truth videos.

### 4.2 Camera Motion Transfer

**Qualitative comparison with the state-of-the-arts.** To validate the effectiveness of our model, we compare our model with the state-of-the-art camera motion control methods on four types of basic camera motions: 1) zoom in, 2) zoom out, 3) pan left, and 4) pan right (in #Suppl). We compare with two motion control methods: 1) AnimateDiff [18] employs the temporal LoRA [22] module to learn the motions from given videos with target camera motions. We train motion LoRA modules on AnimateDiff with one-shot and few-shot data, and compare them with our model. 2) Moreover, we

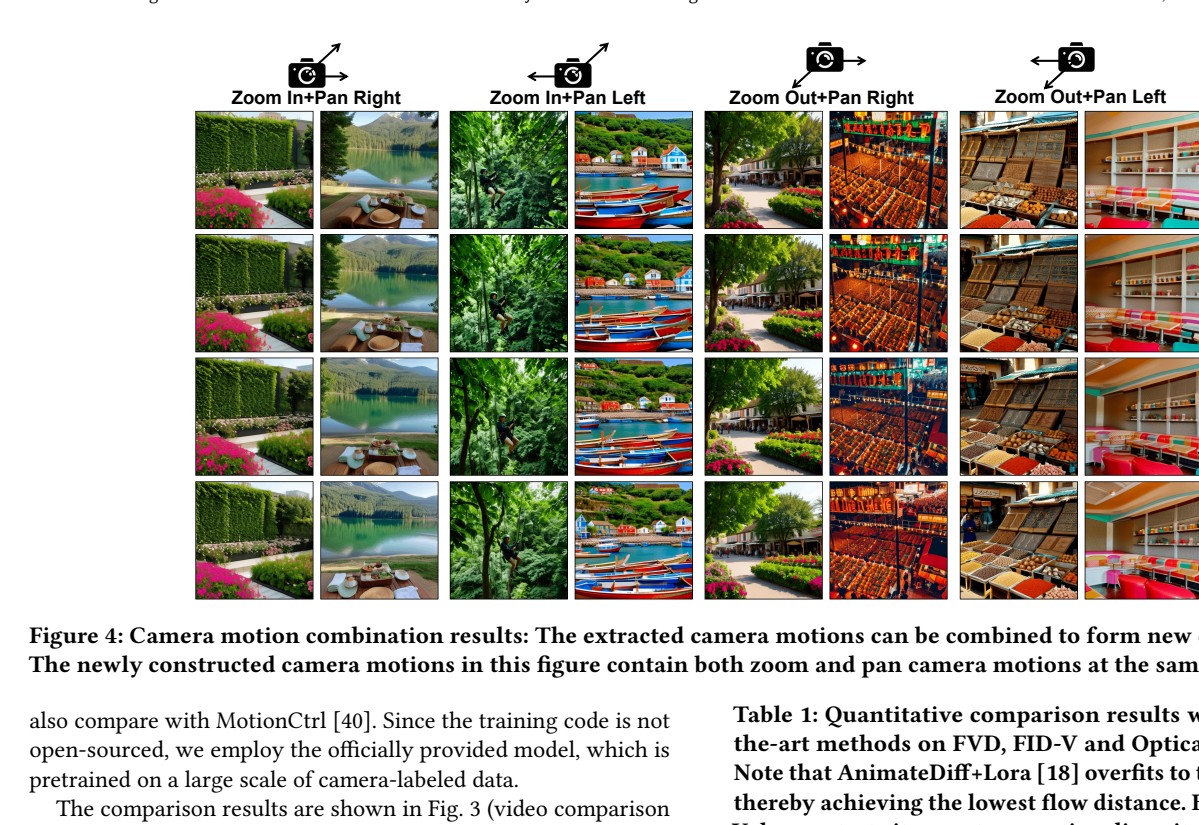

**Figure 4: Camera motion combination results: The extracted camera motions can be combined to form new camera motions. The newly constructed camera motions in this figure contain both zoom and pan camera motions at the same time.**

also compare with MotionCtrl [40]. Since the training code is not open-sourced, we employ the officially provided model, which is pretrained on a large scale of camera-labeled data.

The comparison results are shown in Fig. 3 (video comparison results are provided in #Suppl). It can be seen that in one-shot condition, AnimateDiff tends to overfit to the given video; while in the few-shot condition, AnimateDiff tends to mix the features of the training videos, which cannot generate correct videos corresponding to the given prompts. MotionCtrl can generate videos that better align with the prompts, but may cause shape distortions and logical inconsistencies when controlling camera motion. In contrast, our model can generate high-quality and diverse videos with only one-shot or few-shot data, without the need for training.

**Quantitative comparison.** We also compare with these models quantitatively, using FVD, FID-V, and Optical Flow distance to evaluate the generation quality, diversity, and camera motion accuracy. For each method, we generate 1,000 videos for each type of camera motion and compute FVD and FID-V with 1,000 collected high-quality videos. We also compute the average Optical Flow Distance between the generated videos and given videos. The results are shown in Tab. 1, where our model achieves the best FID-V and FVD, demonstrating superior generation quality and diversity. Since AnimateDiff overfits to the training data, it get a lower Flow distance, but suffers from the worst generation diversity. In summary, our model achieves the best FVD and FID-V, while also ensuring a good camera transfer accuracy compared to MotionCtrl.

## 4.3 Flexible Motion Control

**Motion combination.** In this section, we evaluate the additivity of our disentangled camera motion attention maps. We employ the extracted camera motions including zoom in, zoom out, pan left and pan right in Sec. 4.2 and combine two of them into a new camera motion by Eq.(4). The results are shown in Fig. 4. It can be seen that when combining the zooming motions with the panning motions,

**Table 1: Quantitative comparison results with the state-of-the-art methods on FVD, FID-V and Optical Flow Distance. Note that AnimateDiff+Lora [18] overfits to the training data, thereby achieving the lowest flow distance. But FVD and FID-V demonstrate its worst generation diversity. In contrast, our model achieves the best FVD and FID-V, while also ensuring a good camera transfer accuracy compared to MotionCtrl [40].**

| Data and Method | | Pan Right | | | Zoom In | | |
|---|---|---|---|---|---|---|---|
| Data Scale | Method | FID-V ↓ | FVD ↓ | Flow Dis ↓ | FID-V ↓ | FVD ↓ | Flow Dis ↓ |
| One shot | AnimateDiff | 382.40 | 4956.42 | **19.76** | 482.58 | 6322.46 | **6.91** |
| | **COMD (Ours)** | **54.45** | **921.95** | 37.92 | **61.45** | **863.24** | 12.11 |
| Large Scale | MotionCtrl | 95.83 | 1207.52 | 38.18 | 80.58 | 935.08 | 13.12 |

**(a) Comparison results on one-shot camera motion control.**

| Data and Method | | Pan Right | | | Zoom In | | |
|---|---|---|---|---|---|---|---|
| Data Scale | Method | FID-V ↓ | FVD ↓ | Flow Dis ↓ | FID-V ↓ | FVD ↓ | Flow Dis ↓ |
| Few shot | AnimateDiff | 268.29 | 4629.08 | **14.76** | 251.44 | 3975.41 | **3.12** |
| | **COMD (Ours)** | **61.38** | **1092.09** | 38.94 | **52.90** | **910.76** | 5.10 |
| Large Scale | MotionCtrl | 98.04 | 1196.54 | 55.25 | 80.12 | 928.41 | 7.88 |

**(b) Comparison results on few-shot camera motion control.**

the camera zooms and pans at the same time, which demonstrates that our model can successfully combine different kinds of camera motions together while ensuring generation quality.

**More professional camera motions.** In this section, we show more professional camera motions in the real film industry, including variable-speed zoom and dolly zoom. For variable-speed zoom, where the camera firstly zooms in fast and then zooms in slowly, we crop a video clip from films with this kind of motion, and achieve this motion control by one-shot camera motion disentanglement (Sec. 3.2). For dolly zoom, where the camera in the background region zooms while the camera in the foreground fixes, we employ the local content-preserving camera motion transfer method (Sec. 3.4) to realize it. The results are shown in Fig. 5. It can be seen that our model transfers the variable-speed zoom motion in the

Anonymous Authors

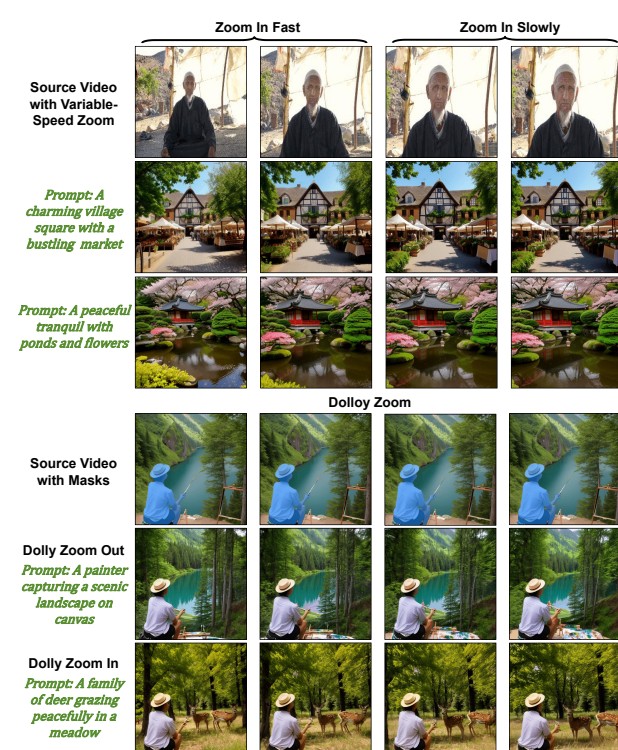

Figure 5: Camera motion control results on professional camera motions, including variable-speed zoom and dolly zoom.

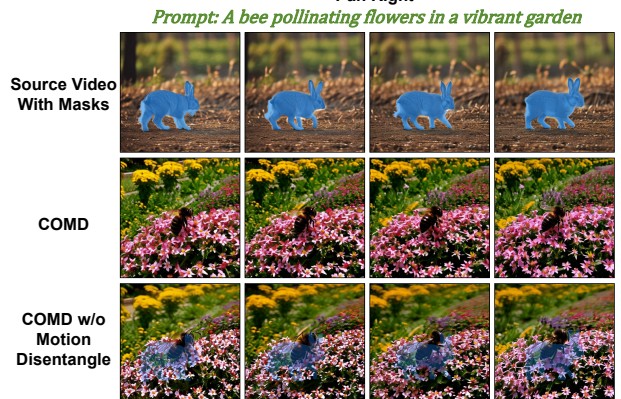

Figure 6: Ablation study on one-shot camera motion disentanglement. The model without motion disentanglement generated artifacts in the region of the moving rabbit.

given video well, and achieves good generation results in both dolly zoom in and dolly zoom out motion controls.

## 4.4 Ablation Study

**Ablation on one-shot camera motion disentanglement.** We first validate the effectiveness of our one-shot camera motion disentanglement method. We compare our model with the ablated version that directly transfers the temporal attention map from the source video to the target video, which does not disentangle the camera and object motions. The results are shown in Fig. 6. It

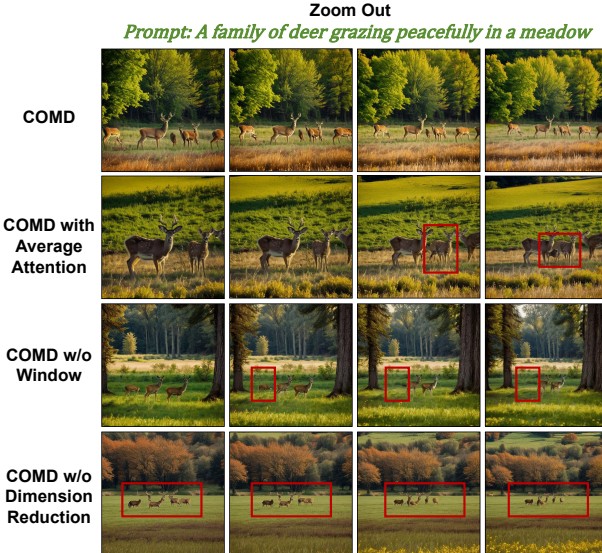

Figure 7: Ablation study on few-shot camera motion disentanglement. All the ablated models generate videos with unnatural movements shown in the red boxes which are caused by the inaccurate extracted camera motions.

can be seen that when transferring the pan right camera motion entangled with the object motion of the moving rabbit, the model without motion disentanglement tends to generate artifacts in the region of the rabbit, which is clearer in the video of #Suppl.

**Ablation on few-shot camera motion disentanglement.** We then validate the effectiveness of our few-shot camera motion disentanglement method. We experiment on three ablated versions on zoom-out camera motion: 1) COMD with Average Attention: the model without DBSCAN clustering and directly averages the camera motions from all the videos; 2) COMD w/o Window: the model without the window-based clustering, which only uses the $m$ pixels at the same location for clustering; and 3) COMD w/o Dimension Reduction: the model without t-SNE to reduce the dimension. The comparison results are shown in Fig. 7. It can be seen that all the ablated models generate unnatural movements shown in the red boxes where certain objects abruptly appear or vanish, or suffer from shape distortions. In contrast, our model achieves the highest generation quality and transfers the camera motions correctly.

## 5 CONCLUSION

In this paper, we propose COMD, a training-free camera motion transfer method based on camera-motion disentanglement. We find that the temporal attention map in the video diffusion model is composed of both camera motion and object motion. We then propose two methods to disentangle the camera motions from object motions for a single or several videos. Moreover, with the extracted camera motions, we further propose a camera motion combination method to enable our model a more flexible and controllable camera control. Extensive experiments demonstrate the superior camera motion transfer ability of our model and show our great potential in controllable video generation.

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
