# OpenReview forum: "COMD: Training-free Video Motion Transfer With Camera-Object Motion Disentanglement"
_acmmm.org/ACMMM/2024/Conference — MM2024 Poster_

### Official Review · Reviewer_oED5 · 2024-05-11

**Rating:** 4
**Confidence:** 4

**Summary:**

This paper proposes COMD, a training-free approach designed to reduce training costs and enable flexible camera control. The method allows for the decoupling of camera motion from object motion and transfers the extracted camera motion to new videos. The primary tasks this method can perform include: (1) implementing a single camera motion (Zoom In), or combining multiple camera motions (Zoom In + Pan Right); (2) keeping the foreground object stationary while enlarging the background (Dolly Zoom In + Object Still).

**Strengths:**

1. The proposed COMD requires no training and utilizes only a single or a few videos, without the need for extensive data, to transfer camera motion from a reference video to a new video. This approach reduces the demand for computational resources and is very data-efficient.
2. The strategy of transforming the camera-motion estimation problem into a Poisson blending problem is interesting and reasonable.
3. The experimental evaluation is reasonably conducted, showing superior performance compared to comparison methods. The ablation studies also demonstrate the effectiveness of the individual components.
4. The paper is well-written with a highly readable style and a logical structure.

**Limitations:**

1. The paper is primarily based on an important observation: temporal attention maps contain information about the motion in videos, which consists of two types: camera motions and object motions. However, it is unclear how the authors derived this observation from temporal self-attention. In fact, as we know, by applying Max-normalization to spatial cross-attention, we obtain spatial attention maps, each corresponding to a subject (entity token in a prompt), allowing us to clearly see the shape associated with the subject. Please describe the overall process of how this observation is derived from the floating temporal self-attention matrix and how it can be visualized.
2. In line 462 and Figure 2, how are the masks from t frames merged into a single mask?
3. In the Source Video & Swap Attention Map experiments, the overall performance is satisfactory, but the preservation of identity seems to be suboptimal, likely due to deformation caused by camera motion. I suspect this issue arises because the method has certain limitations in predicting camera motions within the foreground region. Please explain this phenomenon.

**Suitability:**

3

---

### Official Review · Reviewer_qqKp · 2024-05-17

**Rating:** 3
**Confidence:** 3

**Summary:**

This paper proposes a training-free camera motion transfer framework, named COMD. COMD consists of two workflows for one-shot camera-object motion disentanglement and few-shot camera-object motion disentanglement. One-shot way is conducted by separating the moving objects and the background regions and estimating the camera motion in the moving objects region by solving a Poisson equation. Few-shot way includes a window-based clustering method. Extensive experiments demonstrate COMD is effective.

**Strengths:**

1. The idea on temporal attention map seems effective, which decompose the temporal attention maps of a single video into camera motion and object motion.
2. The motion combination makes this method more practical.
3. Experimental results are good visually.

**Limitations:**

1. "while when t is small, the denoising has almost been completed and the overall motion has already been determined, thus the motion information in the temporal attention map at small t is not sufficient." So what about using temporal attention maps of a small t (i.e., determined overall motion) for a medium timestep t of target video?
2. "the values of the attention map on both sides of the mask boundary are almost the same." This assumption is a bit counter-intuitive, since there is typically more variation at the boundaries (e.g., Canny edge detection).
3. This paper only compares with two competing methods.
4. No limitations are introduced.

Writing issues:
1. Section 3 is a little verbose. Line264-388 are introduced in Section 1. L400-439 are similar to L278-293.
2. Line397-398 should be "predict object motion in the foreground".
3. For Attn_m=Attn*(1-M), i suggest to change Attn_m as Attn_bg.
4. "m" is used for denoting mask and videos.

(If authors could solve my concerns, i may change my rating.)

**Suitability:**

3

---

### Official Review · Reviewer_qwHr · 2024-05-19

**Rating:** 3
**Confidence:** 3

**Summary:**

In this paper, the authors propose a Training-free camera motion transfer model in video, called COMD, to control camera movement. Specifically, two methods are proposed to disentangle the camera motions and object motions in temporal attention maps. 1) Decomposing camera and object motion and predicting camera motion by Poisson equation in one-shot camera motion disentangle; 2) A window-based clustering method is proposed to extract common camera motions from several videos with similar camera motions. Finally, a camera motion combination method is proposed to achieve flexible camera control by combining different kinds of camera motions. Numerous experiments demonstrate the superiority of COMD.

**Strengths:**

1. A one-shot camera-object motion disentanglement method is proposed to separate object and background regions. The camera motion of the moving objects region is estimated by solving the Poisson equation.
2. A few-shot camera-object motion disentanglement method is proposed. It can be used in other camera motion control video generation.

**Limitations:**

1. In the article, lines 266-270 mention “We first observe that the temporal attention maps in diffusion-based video generation models contain the information of videos motions, and find that the motion are composed of two motion types, camera motions and object motions.”. Please further explain the information of video motions in temporal attention maps, and the two motion types that are subsequently separated, camera motions and object motions.

2. Does the method work if there is a noticeable change in the background, such as a travelling cyclist passing by quickly?

3. In the article, lines 266-270 mention “Different from the spatial attention maps (in spatial attention modules), which model different spatial relationships in different timesteps, the temporal attention maps model the temporal motion of the video, and we find they are similar in different timesteps”. Please prove that temporal attention maps are similar in different timesteps.

4. In the paper comparing the methods AnimateDiff [1] and MotionCtrl [2], experiments with a variety of complex camera movements were performed. However, in the experiments in this article, there are only four basic motions and no more complex motions such as scrolling and tilting. It reduces the innovation of this article.

5. In line 23 of the supplementary material, it says “We conduct experiments based on AnimateDiff-v2”. Are the weights of AnimateDiff-v2 also loaded in the COMD? Since camera motion control is already implemented in AnimateDiff, isn't “Training-free” a bit of a stretch if COMD also uses AnimateDiff-v2 weights? Please explain it.

[1] Guo Y, Yang C, Rao A, et al. Animatediff: Animate your personalized text-to-image diffusion models without specific tuning[J]. arXiv preprint arXiv:2307.04725, 2023.

[2] Wang Z, Yuan Z, Wang X, et al. Motionctrl: A unified and flexible motion controller for video generation[J]. arXiv preprint arXiv:2312.03641, 2023.

**Suitability:**

3

---

### Official Review · Reviewer_i7ZR · 2024-05-22

**Rating:** 3
**Confidence:** 3

**Summary:**

This paper proposes a training-free video motion transfer model, which disentangles camera motions and object motions in source videos and transfers the extracted camera motions to new videos. A one-shot camera motion disentanglement method, a few-shot camera motion disentanglement method, and a motion combination method is proposed.  Extensive experiments demonstrate that the method can effectively decouple camera-object motion and apply the decoupled camera motion to a wide range of video generation tasks, achieving flexible and diverse camera motion control.

**Strengths:**

1. A one-shot camera motion disentanglement method is proposed to extract camera motion from a single source video, which separates
the moving objects from the background and estimates the camera motion in the moving objects region based on the motion in the
background by solving a Poisson equation.
2. A few-shot camera motion disentanglement method is proposed to extract the common camera motion from multiple videos with similar camera motions.
3. A motion combination method is proposed to combine different types of camera motions together.
4. The writing is good.

**Limitations:**

1. The authors do not provide the visualizations of the motions. Make it unclear to estimate the motion disentanglement method.
2. The experimental sections lack the comparison of running times of different methods. For example, GPU/CPU time.
3. No quantitative ablation study results provided.

**Suitability:**

3

---

### Meta-Review · Area_Chair_pMvh · 2024-07-01

**Recommendation:** Accept (Poster)
**Confidence:** 4

**Metareview:**

The paper introduces COMD, a training-free framework for video motion transfer that disentangles camera and object motions from source videos and transfers extracted camera motions to new videos. It proposes innovative methods for one-shot and few-shot camera motion disentanglement, alongside a motion combination approach. Despite concerns about lacking visualizations of the motions and detailed runtime comparisons, the extensive experiments demonstrate effective motion control and competitive performance compared to existing methods. Reviewers note the clear writing and the method's potential for diverse applications in multimedia processing.